# The TGF-β Receptor Gene *Saxophone* Influences Larval-Pupal-Adult Development in *Tribolium castaneum*

**DOI:** 10.3390/molecules27186017

**Published:** 2022-09-15

**Authors:** Jingjing Li, Letong Yin, Jingxiu Bi, David Stanley, Qili Feng, Qisheng Song

**Affiliations:** 1Division of Plant Science and Technology, University of Missouri, Columbia, MO 65211, USA; 2Institution of Quality Standard and Testing Technology for Agro-Product, Shandong Academy of Agricultural Science, Jinan 250100, China; 3Biological Control of Insect Research Laboratory, United States Department of Agriculture-Agricultural Research Station (USDA/ARS), Columbia, MO 65203, USA; 4Guangzhou Key Laboratory of Insect Development Regulation and Application Research, Institute of Insect Science and Technology, School of Life Sciences, South China Normal University, Guangzhou 510631, China

**Keywords:** *saxophone*, TGF-β receptor, *Tribolium castaneum*, development, RNAi, gene expression

## Abstract

The transforming growth factor-β (TGF-β) superfamily encodes a large group of proteins, including TGF-β isoforms, bone morphogenetic proteins and activins that act through conserved cell-surface receptors and signaling co-receptors. TGF-β signaling in insects controls physiological events, including growth, development, diapause, caste determination and metamorphosis. In this study, we used the red flour beetle, *Tribolium castaneum,* as a model species to investigate the role of the type I TGF-β receptor, *saxophone* (*Sax*), in mediating development. Developmental and tissue-specific expression profiles indicated *Sax* is constitutively expressed during development with lower expression in 19- and 20-day (6th instar) larvae. RNAi knockdown of *Sax* in 19-day larvae prolonged developmental duration from larvae to pupae and significantly decreased pupation and adult eclosion in a dose-dependent manner. At 50 ng dsSax/larva, Sax knockdown led to an 84.4% pupation rate and 46.3% adult emergence rate. At 100 ng and 200 ng dsSax/larva, pupation was down to 75.6% and 50%, respectively, with 0% adult emergence following treatments with both doses. These phenotypes were similar to those following knockdowns of 20-hydroxyecdysone (20E) receptor genes, *ecdysone receptor* (*EcR*) or *ultraspiracle*  *protein* (*USP*). Expression of 20E biosynthesis genes *disembodied* and *spookier*, 20E receptor genes *EcR* and *USP*, and 20E downstream genes *BrC* and *E75*, were suppressed after the *Sax* knockdown. Topical application of 20E on larvae treated with dsSax partially rescued the dsSax-driven defects. We can infer that the TGF-β receptor gene *Sax* influences larval-pupal-adult development via 20E signaling in *T. castaneum*.

## 1. Introduction

The transforming growth factor beta (TGF-β) superfamily encodes multifunctional growth factors. It is expressed in many vertebrates and invertebrates, including one of the earliest multicellular animals (*Trichoplax adhaerens*) [1]. Based on their structural and biological similarities [2], the superfamily is divided into two major branches, the bone morphogenetic proteins (BMP)/growth differentiation factor branch and the TGF-β/Activin/Nodal branch.

TGF-β signals in insects control a variety of physiological processes, including growth, development, diapause, caste determination and metamorphosis. They are responsible for development and immunity in vertebrates [3]. In the tobacco hornworm, *Manduca sexta,* and the red flour beetle, *Tribolium castaneum*, the TGF-β/Activin ligand Myoglianin (Myo) mediates larval growth and development [4]. In the cotton bollworm, *Helicoverpa armigera*, TGF-β and BMP signals regulate development and diapause through a cascade of transcription factors, suppressor of mothers against decapentaplegic family member 1 (Smad1), pituitary-octamer-unc (POU) and transcription factor A of mitochondria (TFAM) [5]. TGF-β signaling acts in the juvenile hormone (JH) and 20-hydroxyecdysone (20E) signaling during soldier differentiation in the Nevada termite, *Zootermopsis nevadensis* [6]. In the two-spotted cricket, *Gryllus bimaculatus*, the TGF-β ligands, Myo and decapentaplegic (Dpp)/glass bottom boat (Gbb), regulate JH synthesis via JH acid O-methyltransferase. Loss of Myo function inhibits metamorphosis [7]. In *Drosophila* TGF-β signals regulate several developmental processes including embryonic and imaginal disc patterning [8], among others. Loss of Activin signaling leads to low 20E levels via down-regulating the 20E biosynthetic genes *disembodied* (*Dib*) and *spookier* (*Spok*), which leads to developmental arrest [9]. Activin signaling influences neuronal remodeling by regulating expressions of the 20E receptor, *ecdysone receptor* (*EcR*) B1 isoform during larva-adult metamorphosis [10]. Dpp signaling regulates JH biosynthesis by activating the expression of *JH acid methyltransferase* (*JHAMT*) in *Drosophila* larvae, and the synthesis of amnioserosa, a major source of 20E, during embryogenesis [11,12].

TGF-β signals work through heterotetrametric complexes of type I and type II dual-specificity kinase receptors. These receptors are expressed in all cell types and act in mediating multiple responses. The structural features of these receptors are highly conserved [13]. The type I and type II receptors form distinct subgroups in the serine/threonine kinase receptor family based on the sequences of the kinase domains and the presence of a highly conserved glycine-serine region called the GS domain. The phosphorylation of the GS region in TGF-β type I receptors by type II receptors enhances the affinity of type I receptors for downstream transcription factors [14]. Saxophone (Sax) is a type I TGF-β receptor, which was first characterized in *Drosophila melanogaster* [15]. Sax, along with Thickveins (Tkv), are essential mediators of BMP signaling. In *Drosophila*, a mutation that abolished Sax expression leads to dysfunctions similar to those associated with partial loss of Dpp signaling, in that dorsal cell fates are missing, and the embryos are partially ventralized [16]. The Sax mutant leads to shifts in cell fate along the anterior-posterior axis, which is essential for the proper growth and patterning of appendages [17]. Similar results occurred in work with the turnip sawfly, *Athalia rosae* [18].

Nonetheless, the roles of Sax in regulating insect growth and development remain unclear. Since TGF-β operates in virtually all animals, from sponges to humans and fish [19], we hypothesized that the insect TGF-β receptor, Sax, operates in insect development. In this paper, we report on the outcomes of experiments designed to test our hypothesis on the red flour beetle, *T**ribolium castaneum*, a serious international pest of stored grain and a widely used model insect [20].

## 2. Materials and Methods

### 2.1. Insect Rearing

The *T. castaneum* Georgia-1 strain was used for all experiments. Insects were reared on organic wheat flour which contained 10% yeast at 28 ± 1 °C under standard conditions [21].

### 2.2. Sequence Alignment, Protein Modeling and Phylogenetic Analysis

The predicted Sax encoding gene sequences from 6 species were downloaded from the National Center for Biotechnology Information (https://www.ncbi.nlm.nih.gov/, accessed 20 July 2022). Multiple sequence alignment was performed using ClustalW2 (https://www.ebi.ac.uk/Tools/msa/clustalw2/, accessed 1 July 2022). This was subsequently visualized and annotated using Jalview software (Version: 2.11.1.4) [22]. Motif search and analysis were performed through MEME (Version: 5.4.1) (https://meme-suite.org/meme/tools/meme, accessed 1 July 2022). The receptor protein kinase structures were first modeled with SWISS-MODEL [23], and then labeled with PyMOL for special motifs [24]. A phylogenetic tree was constructed based on amino acid sequence alignment by the neighbor-joining algorithm embedded in the MEGA XI program [25]. The tree was drawn to scale, with branch lengths in the same units as those of the evolutionary distances used to infer the phylogenetic tree and was modified using ITOL (https://itol.embl.de/upload.cgi, accessed 25 July 2022).

### 2.3. Developmental and Tissue-Specific Sample Preparation

Samples were collected from pools of multiple individuals in the developmental stages: 1- and 5-day eggs (~0.05 g each), 1-, 5- and 10-day larvae (~0.05 g each), 15-, 18-, 19- and 20-day larvae (three individuals each), 1–5-day pupae (three individuals each), and 0.5 and 1 h adults (three individuals each). Adults with untanned cuticles were staged as 0 h after emergence. For analysis of tissue-specific expression, the central nervous system (CNS), fat body (FB), gut, epidermis, hemocyte and Malpighian tubules were isolated from pools of approximately 90 19-day larvae. Thirty pairs of eyes were dissected from dsSax-treated and dsGFP control larvae 48 h post-injection (PI), respectively. Similarly, 30 pairs of genitalia were dissected from 1-day female pupae and male pupae of dsSax-treated and control groups. Three biological replicates were conducted for each treatment.

### 2.4. RNA Extraction, cDNA Synthesis and qRT-PCR

Total RNAs were extracted from the whole bodies or selected tissues of staged beetles using TRIzol reagent (Thermo Fisher Scientific Inc., Waltham, MA, USA). DNA was eliminated from the total RNA samples using DNase I (Thermo Fisher Scientific Inc., USA). Reverse transcription was performed using 1 μg total RNA. A High-Capacity cDNA Reverse Transcription Kit (Thermo Fisher Scientific Inc., USA) with RNase inhibitor in a 20 μL reaction volume was used for cDNA synthesis following the manufacturer’s instructions. Quantitative real-time PCR (qRT-PCR) was performed to estimate *T. castaneum Sax* (*TcSax*) expression patterns using a QuantStudio 3 Real-Time PCR System (Thermo FAisher Scientific Inc., USA). qRT-PCR reaction components were 1 μL of cDNA (100 ng/μL), 1 μL each of forward and reverse sequence-specific primers (10 pmol/μL), 3 μL of H_2_O and 5 μL of iTaq™ Universal SYBR^®^ Green Supermix (Biorad Laboratories, Hercules, CA, USA). The PCR system was programmed for 95 °C for 3 min, 45 cycles of 95 °C for 10 s, 60 °C for 20 s, 72 °C for 30 s and 65–95 °C at 0.5 °C increments for 2–5 s. Relative mRNA expression levels were quantified and normalized using a stably expressed internal control (ribosomal protein S3, Tcrp3 mRNA) [26]. The primer sequences for target genes are listed in Appendix A. Three biological replicates with 3 technical repeats each were performed. The relative expression levels of genes were calculated according to the 2^−ΔΔCT^ method [27].

### 2.5. Double-Stranded RNA (dsRNA) Synthesis and RNAi Assay

The templates for dsRNA (dsGFP, dsSax, dEcR and dsUSP) synthesis were obtained by PCR for each gene using gene-specific primers containing the T7 polymerase promotor sequence at their 5′ ends (Appendix A). The resulting cDNA was used as the template. dsRNA was synthesized using purified PCR products and the HiScribe™ T7 Quick High Yield RNA Synthesis Kit (E2050, New England Biolabs Inc., Ipswich, MA, USA) following the methods described in the instruction manual. Synthesized dsRNA was purified using a phenol/chloroform extraction and isopropanol precipitation method and dissolved in diethylpyrocarbonate-treated water [28]. Afterwards, the formation of dsRNA was monitored by determining the molecular size using agarose gel electrophoresis. The concentration of dsRNA was measured using a Nanodrop 2000 spectrophotometer (Thermo Fisher Scientific Inc., USA) at 260 nm. The dsRNA was injected into the dorsal side between the 8th and 9th abdominal segments of each 19-day larva using the Nanoject II Auto-Nanoliter Injector (Drummond Scientific Co., Broomall, PA, USA) fitted with a 3.5-inch glass capillary tube pulled by a needle puller (Model P-2000, Sutter Instruments Co., Novato, CA, USA). Larvae were maintained under standard conditions. Each larva was injected with 50 nL dsRNA containing the indicated concentrations (0.5, 1, 2, 4 ng/nL). The dsGFP was used as a control, as previously described [29].

### 2.6. Effect of RNAi on Phenotypes

The *T. castaneum* phenotypes were photographed using a Leica M205 C stereomicroscope with a digital camera. Photos were taken from the ventral and dorsal sides of pupae and adults to record any visible phenotypic changes during the development of injected beetles. Pupae were distinguished by sex, based on the structural differences in the genital papillae.

### 2.7. Effect of RNAi on Larval Duration, Pupation Rate, and Adult Eclosion Rate

The *T. castaneum* were divided into three groups (30 larvae per group) to determine larval developmental duration, pupation rate and adult eclosion rate after RNAi treatments. dsGFP was used as controls. The experiments were repeated three times. For the larval developmental duration, the number of pupae was recorded daily from 60 larvae injected with dsRNA.

### 2.8. Topical 20E Application onto the dsRNA-Treated Larvae

To evaluate 20E rescue responses, technical grade 20E (Adipogen Corp., San Diego, CA, USA) was dissolved in acetone at the indicated concentrations (0, 0.2, 2 and 4 µg/µL), and was topically applied (0.5 µL/larva) onto the dorsal side surface of 19-day larvae at 24 h post 100 ng/larva of dsSax treatment. The applications were performed on individual larvae using a repeating syringe dispenser (PB600-1 Hamilton syringe with a 25-µL syringe). The larval duration, pupation rate and adult emergence rate were recorded.

### 2.9. Statistical Analysis

Gene expression levels and the other parameters of the dsSax-treated versus control groups were compared by one-way analysis of variance (ANOVA) in combination with Student’s *t*-test using the graphic software Prism (Graph Pad Software, v8.1.2, San Diego, CA, USA). The larval duration, pupation rate and adult emergence rate between the dsSax-treated and control groups or 20E rescue groups were compared by an unpaired *t*-test. All data obtained were presented as the mean ± standard error of mean (SEM) from three or more independent experiments. A *p*-value < 0.05 was regarded as statistically significant.

## 3. Results

### 3.1. Phylogenetic Analysis of Sax

TcSax was compared with Sax from *D. melanogaster* (NP_523652.2), Activin type I receptors (ActivinRI) from *Z. nevadensis* (KDR18456.1), *Aedes aegypti* (XP_021696047.1) and *M. sexta* (XP_037301442.1), and the BMP type I receptor (BMPRI) from *H. armigera* (AWV63129.1). TcSax has high similarity to other TGF-β type I receptors. TcSax contains a complete GS domain and a conserved juxtamembrane region located immediately at the amino-terminal of their kinase domain. TcSax shares the highest identity (66.5%) with ActivinRI from *Z. nevadensis*, followed by sequences from *Ae. aegypti* (65.3%), Sax from *D. melanogaster* (60.4%), ActivinRI from *M. sexta* (48.9%) and BMPRI from *H. armigera* (45%) (Figure 1A). Motif analysis confirmed that the highly-conserved GS motif occurs in a six amino acid sequence described in Figure 1A (Figure 1B). Protein kinase structure prediction and GS motif labeling indicate TcSax and DmSax have similar molecular structures (Figure 1C).

A phylogenetic tree, based on the amino acid sequences of Sax and TGF-β superfamily type I receptors (TGF-βRIs) from 111 representative species of Arthropods and a Chordate, shows that TcSax is conserved with TGF-βRIs from other insect species, particularly within insect orders. In general, insect TGF-βRIs and some crustacean TGF-βRIs are closer to each other compared to vertebrates (Figure 2).

### 3.2. Developmental and Tissue-Specific Expression of Sax

*Sax* is expressed through all developmental stages, with high expression in eggs, especially 5-day eggs, and low expression in 19- and 20-day larvae (6th instar), late pupae and 1 h adults. (Figure 3A). *Sax* is constitutively expressed in the indicated tissues and organs prepared from 19-day larvae (Figure 3B).

### 3.3. Effects of Sax RNAi on Larval-Pupal Development

As the dsSax injection dose increased from 25 to 200 ng per larva, the RNAi efficiency gradually increased from 29.5% to 83.2% at 72 h PI (Figure 4A). To examine the dsSax RNAi efficiency, samples were taken from larvae injected with 100 ng/larva at 12, 24, 48, 72 and 96 h after dsSax injection. Figure 4B shows *Sax* expression was significantly decreased from 24 to 96 h PI, and the RNAi efficiency increased from 50% at 48 h to 75.3% at 96 h compared to the control group.

We examined 60 larvae injected individually with 100 ng dsSax or 100 ng dsGFP for controls and recorded the numbers of newly emerged pupae daily. Sax RNAi treatments led to extended larval development time from 3.7 to 4.5 days, with the most prolonged duration of 7 days (Figure 4C).

Some dsSax-treated larvae did not pupate, but developed abnormal eyes, with their development arrested during the quiescent stage (Figure 4D).

### 3.4. Effects of Sax RNAi on Pupation and Adult Emergence

Pupation and adult eclosion decreased in a dose-dependent manner after injecting dsSax (Figure 5A,B). Compared to controls, the 25 ng dsSax/larva dose had no effect on the pupation and eclosion (Figure 5A,B). At 50 ng dsSax/larva, 15.5% of the dsSax treated larvae did not pupate and 53.7% of the pupae did not emerge as adults (Figure 5A,B). At 100 ng dsSax/larva, 75.5% of the treated larvae were pupated, but none of these pupae became adults (Figure 5A,B). At 200 ng/larva dsSax, only 50% of the larvae were pupated and none of the resulting pupae became adults (Figure 5A,B).

Adults that emerged from larvae treated with 25 ng dsSax/larva did not have visible morphological differences from the controls (Figure 5C). In the injected larvae, the heads and thoraces of adults that emerged from larvae treated with 50 or 100 ng of dsSax had visible adult features, although the abdomens retained juvenile characteristics and they did not complete pupal-adult metamorphosis. After injecting 200 ng dsSax/larva, the pupae featured tanned cuticle and adult coloration with no other adult traits (Figure 5C).

### 3.5. dsSax Treatments Led to Retarded Development

We recorded phenotypes of pupae and newly emerged adults from larvae injected with 100 ng/larva dsSax. Before eclosion, female and male pupae of the control groups had dark coloration, while the dsSax-treated pupae did not develop to the pharate stage, although there was black pigmentation on the pupal wings (Figure 6A). The heads and thorax parts of dsSax-related larvae developed adult traits although the abdomens remained in the pupal shape (Figure 6A and Appendix A). From newly formed pupae to adults, the morphology of pupal genitals in the dsSax-treated group differed from controls. The genitalia of male pupae were enlarged compared to controls (Figure 6B), and the female genitalia appeared abnormal (Figure 6B). These genitalia phenotypes lasted from pupal emergence to adult eclosion (Figure 6A,B). The weights of dsSax-treated female and male pupae did not differ from controls, both means within 0.0028 to 0.0030 g (Appendix A).

### 3.6. Effects of Sax RNAi on Expression of 20E Related Genes

Treating larvae with 100 ng dsEcR or dsUSP led to decreased expression of *EcR* and *USP* at 24 h PI (Appendix A). The dsEcR and dsUSP treatments led to slight increases in the larval period (Appendix A). The treatments led to reduced pupation, down from 100% to 35.9% and 36.3% PI (Appendix A). The dsEcR and dsUSP-treated larvae did not develop into adults (Appendix A). These results mimic results recorded in dsSax-treated larvae. The dsEcR and dsUSP treatments led to altered phenotypic outcomes similar to the results of the dsSax experiments (Appendix A, Figure 4D and Figure 5C). We considered a possible connection between the Sax and the 20E pathways. Larvae were treated with 100 ng dsSax/larva and evaluated for expression of 20E- and JH-related genes. After dsSax treatment, the expressions of the 20E synthesis-related Halloween genes *Dib* and *Spok* were significantly reduced at 24 h and 48 h PI (Figure 7A,B), while the expression of *phantom* and *shadow* did not change (Appendix A). This is consistent with the results in *D. melanogaster* [9]. We speculated that *Dib* and *Spok* are the regulatory targets of Sax signaling. Consequently, the expression of 20E receptor genes *EcR* and *USP* was significantly downregulated by 50% and 45%, respectively, at 24 and 48 h PI, relative to controls (Figure 7C,D). Furthermore, the expression of the 20E downstream genes *BrC* and *E75* was also reduced by about 60% and 50% at 48 h PI (Figure 7E,F). We then detected the expression of *Sax* and 20E pathway-related genes in abnormal eyes dissected from dsSax-treated and normal eyes from control larvae. The results showed that the expression of *Sax* and most 20E pathway genes were significantly down-regulated in the abnormal eyes from dsSax-treated larvae compared with normal eyes in the control group (Appendix A). Similar results were also obtained in the genitalia dissected from 1-day female pupae and male pupae of dsSax-treated larvae (Appendix A), indicating that Sax RNAi might contribute to the abnormal development via the 20E pathway.

dsSax treatments led to no significant changes in expression of the JH synthesis-related gene, *JHAMT*, nor in the JH receptor gene *Tai* at 12 and 48 h PI (Appendix A). The JH receptor gene *Methoprene-tolerant* (*Met*) and downstream gene *Krüppel homolog 1* (*Krh1*) were upregulated by about 2-fold and 4-fold, respectively, at 24 h PI. Expression of these two genes returned to normal by 48 h PI (Appendix A). Expression of the JH hydrolysis enzyme, *JH esterase* (*JHE*), was downregulated at 24 h PI and then upregulated at 48 h PI (Appendix A).

### 3.7. 20E Application Rescues the dsSax Treated Larvae

We performed an *in vivo* hormone rescue experiment by topically applying 20E at 0.1, 1 and 2 μg/larva at 24 h post dsSax (100 ng/larva) treatment. The 1 μg/larva 20E treatments led to decreased larval durations, down from 4.6 days to 3.9 days in dsSax treated larvae (Figure 8A). The 1 μg/larva 20E rescue treatments led to increases in pupation rate from 73.7% in dsSax-treated larvae to 82.9% (Figure 8B). The 20E application also led to adult emergence in 19.3% of larvae (Figure 8C). We found the 0.1 μg/larva 20E treatments led to no significant difference in larval duration and pupation, but the adult emergence rate increased to 4.9% (Figure 8A–C). The 2 μg/larva 20E treatments did not lead to further increases in the biological outcomes (Figure 8A–C). Although most larvae treated with 1 μg/larva 20E did not emerge as adults, the 20E treatment led to more adult features in the pupal-adult intermediates (Figure 8D).

## 4. Discussion

The data presented in this paper strongly support our hypothesis that the insect TGF-β receptor, Sax, operates in development. Several points are germane. First, TcSax is confirmed to be TGF-βRI and it was expressed in all developmental stages and in all six tissues and organs we tested. Second, injections of a dsRNA construct specific to TcSax led to dose-related reductions in gene expression that lasted at least 96 h and the injections led to increased larvae periods. Third, pupation and adult emergence declined following dsSax treatments. Fourth, the dsSax injections led to arrested development and to abnormal phenotypes. Fifth, the dsSax treatments led to reduced expression of genes operating in 20E signaling, including 20E synthesis-related genes *Dib* and *Spok*, 20E receptor genes *EcR* and *USP*, and 20E downstream genes *BrC* and *E75.* Finally, rescue treatments with 20E led to partial reversals of the dsSax influence. Taken together, these points amount to a forceful argument that the TGF-β receptor, TcSax, is a major element in the endocrine-driven development of the red flour beetle and likely many other insect species.

TGF-β signaling acts in embryogenesis, stem cell development and organogenesis [30]. The TGF-β signaling pathway communicates with other signaling pathways to regulate cellular functions. Dysregulation of TGF-β responsiveness and its downstream signaling pathways contributes to adverse consequences, including embryonic lethality and inflammation [31,32]. TGF-βRI is a key component in passing extracellular stimulation to the downstream TGF-β signaling pathway [32]. It also functions in the absence of type II receptors to promote larval development in the nematode *Caenorhabditis elegans* [33]. Animals have evolved reliance on TGF-β signaling during many aspects of development.

In this study, through the alignment of the amino acid sequences of TcSax and TGF-βRIs from other insect species, motif analysis, protein modeling and further phylogenetic tree analysis of TGF-βRIs from the other animals, we confirmed that TcSax is a TGF-βRI. The subsequent expression of TcSax through development, particularly coupled with the data showing that knockdown of TGF-βRI Sax via RNAi affected larval and pupal development and caused failure in larval–pupal and pupal–adult ecdysis in *T. castaneum*. The occurrence of this abnormal phenotype was dose-dependent, indicating that TcSax acts in development.

We found that dsSax treatments led to decreased expression of the 20E signaling-related genes. For insects, 20E and JH signal a wide range of events in growth and development [34]. Insect Halloween genes such as *phantom*, *Dib*, *shadow* and *Spok*, encode P450 enzymes that act in biosynthesis of 20E [35,36]. The 20E exerts its effects through a heterodimeric receptor composed of EcR and USP [37]. The 20E/EcR/USP complex binds to ecdysone response elements to activate early 20E-response genes, such as *BrC* and *E75*. EcR and USP are essential for insect metamorphosis [28]. *BrC* controls pupal commitment and pupal morphogenesis and inhibits adult differentiation in *T. castaneum* [21,38,39]. E75 is involved in pupal–adult cuticle pigmentation and sclerotization in *T. castaneum* [40]. TGF-β signaling triggers the JH and 20E pathway, and possibly others to mediate the larval growth, development and metamorphosis in insects generally [4,5,6,7,10]. Inhibiting expression of these 20E-related genes by dsSax treatment leads to abnormal growth and development. The results indicate that Sax signaling operates through the 20E pathway via influencing the expression of 20E biosynthesis-related genes *Dib* and *Spok*, the 20E receptor genes *EcR* and *USP*, and the 20E downstream genes *BrC* and *E75*.

The topical application of 20E partially reversed the dsSax effects. The rescue effect is somewhat limited because Sax is expressed constitutively and is not limited to the 20E signaling system. We speculate that a detailed analysis of the rescue limitations may lead to the discovery of currently unknown specific signaling mechanisms in insect development.

The JH receptor gene *Met* operates in development and JH titer determines the outcome of molting in *T. castaneum* [41]. *Krh1* represses key genes such as *BrC.* It is involved in adult development and in the expression of steroidogenic enzymes in immature stages [42,43]. Based on our results, dsSax treatments did not influence the expression of the JH synthesis-related gene *JHAMT* nor the receptor gene *Tai* but upregulated the expression of *Met* and *Krh1* at 24 h post dsSax injection. Our interpretation is that while dsSax appears to interact with elements of JH signaling via undiscovered mechanisms, it does not directly influence it.

We propose that the TGF-β receptor, Sax, contributes to larval–pupal–adult development via influencing genes acting in 20E synthesis. Nevertheless, we explored the interaction between the Sax and 20E pathways at the gene expression level. Research into the influence of dsSax on protein activities and the crosstalk between the TGF-β receptor and 20E signaling on insect growth and development is currently underway. Our research contributes to understanding the biological significance of Sax in insect TGF-β signaling and it lays a foundation for a deeper understanding of insect development and metamorphosis.

## Figures and Tables

**Figure 1 molecules-27-06017-f001:**
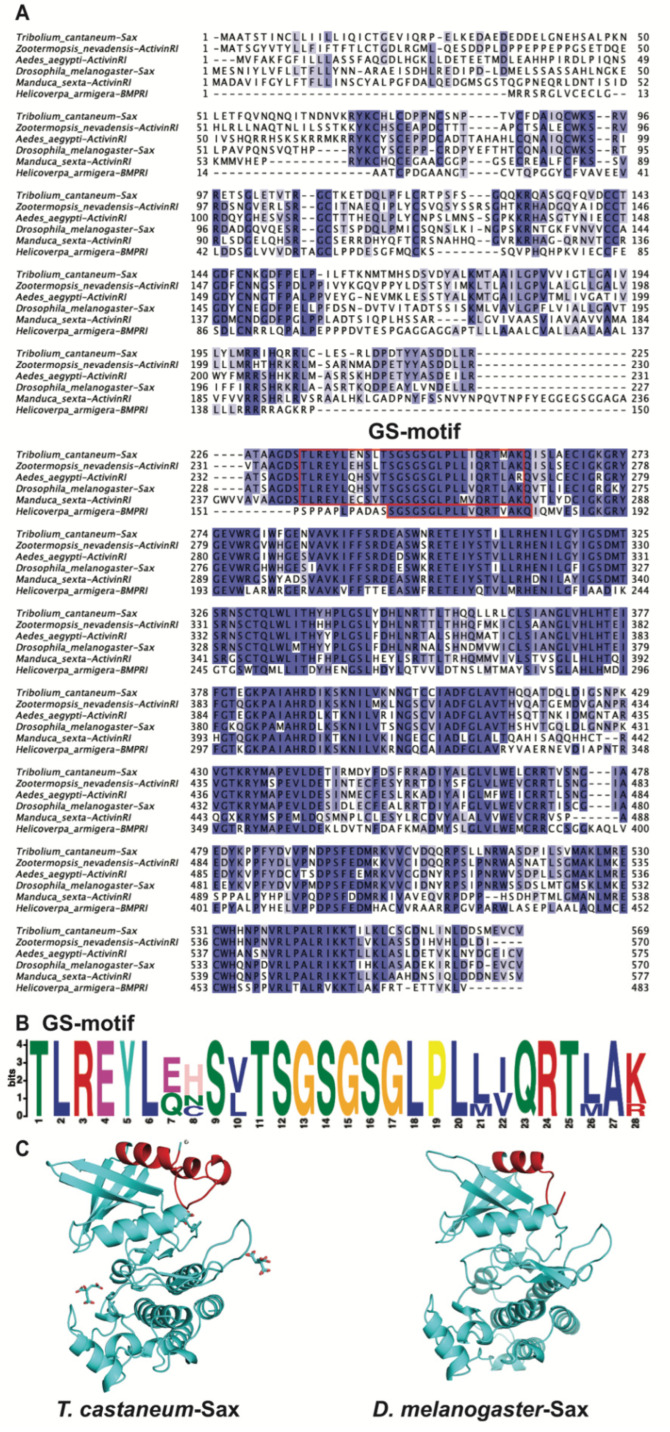
Sequence alignment and protein modeling of TGF-βRIs. (**A**) Multiple-sequence alignment of TGF-βRI amino acid sequences. Conserved amino acid sequences are indicated by a dark blue background. Identical and highly similar amino acid sequences are indicated by a light blue background. Sequences surrounded by red frames represent the GS motif. (**B**) The MEME motif search result of GS motif in the sequences in Figure 1A. (**C**) Molecular model of the three-dimensional structure of Sax kinase domain in *T. castaneum* and *D. melanogaster*. The GS motif is colored red.

**Figure 2 molecules-27-06017-f002:**
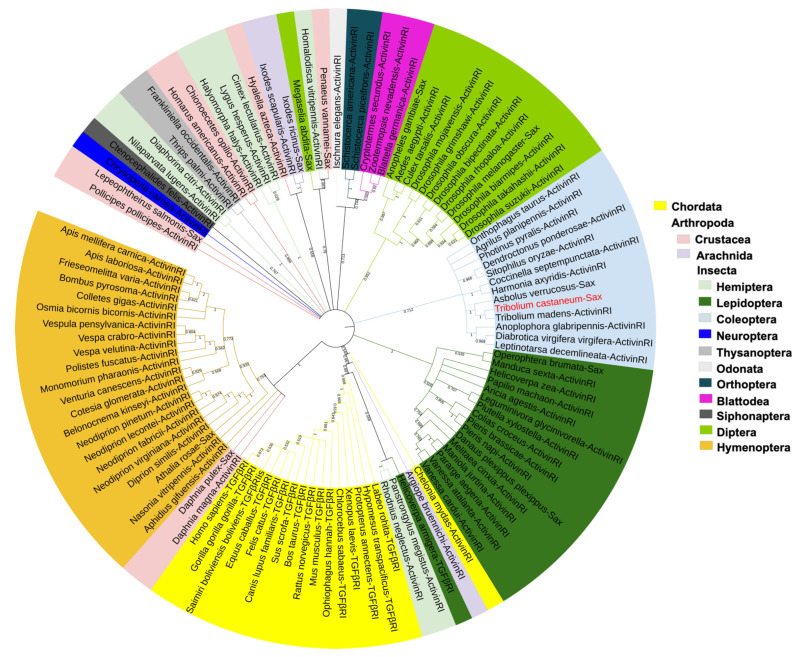
Phylogenetic tree of Sax and other TGF-β superfamily type I receptors (ActivinRI: Activin type I receptor; TGFβRI: TGF-β type I receptor). The phylogenetic tree was constructed using MEGA software XI through the neighbor-joining method with 1000 bootstrap replications. TcSax was marked with red font. The accession numbers of protein sequences used for phylogenetic analysis was shown in Appendix A.

**Figure 3 molecules-27-06017-f003:**
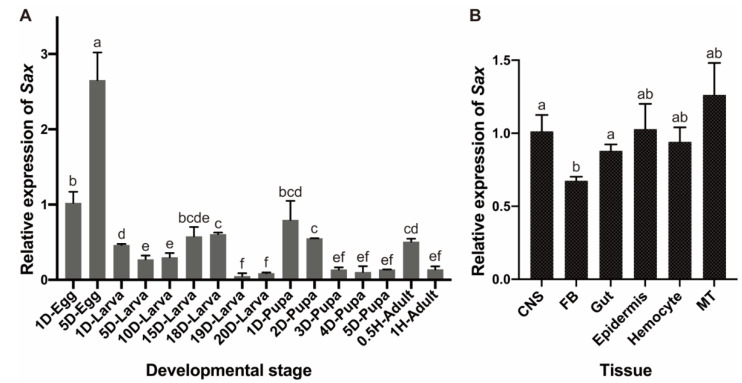
Spatiotemporal expression of *Sax* in *T. castaneum.* (**A**) Developmental expression pattern of *Sax*. The relative expressions of the target transcripts in the 1-day egg served as the calibrator for the developmental expression profiling. (**B**) Tissue-specific expression patterns of *Sax* in central nervous system (CNS), fat body (FB), gut, epidermis, hemocyte and Malpighian tubule (MT) from 19-day larvae. Different letters on the bars indicate that the means ± SEM are significantly different among treatments by *t*-test.

**Figure 4 molecules-27-06017-f004:**
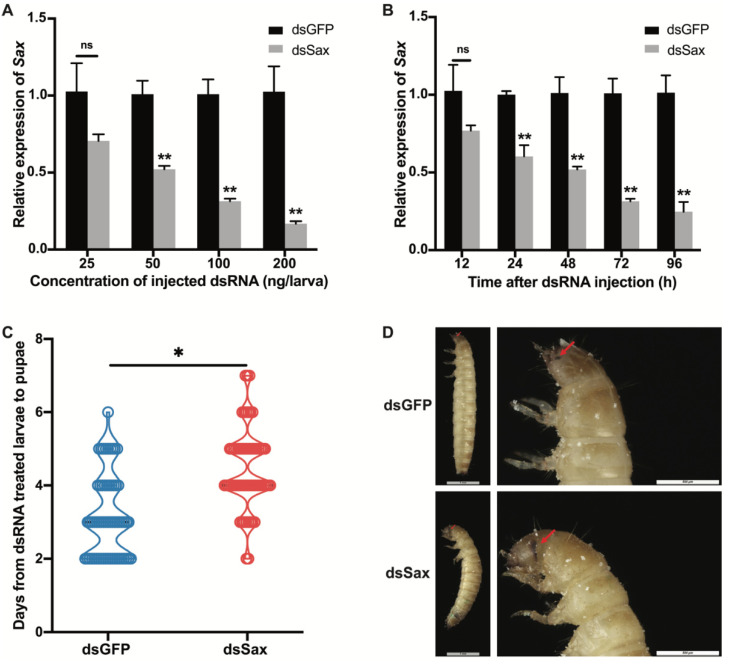
The efficiency of *Sax* RNAi and its effect on larval-pupal development. (**A**) Relative expression levels of *Sax* 72 h after different doses of dsSax injection. (**B**) Relative expression levels of *Sax* 12–96 h post 100 ng/larva of dsSax injection. (**C**) Days from dsRNA treated 19-day larvae to newly formed pupae from the dsGFP and dsSax-treated groups (100 ng/larva). (**D**) Phenotype of larva in the dsGFP (100 ng/larva)-treated groups and larva arrested during the quiescent stage in the dsSax (100 ng/larva)-treated groups. The red arrows indicated the normal eye and abnormal eye development in the *Sax* RNAi larva. NS indicated non-significant differences between the treatment and corresponding control, *p* > 0.05 by *t*-test, asterisks above bars indicate significant differences between the treatment and the control, * *p* < 0.05, ** *p* < 0.01 by *t*-test.

**Figure 5 molecules-27-06017-f005:**
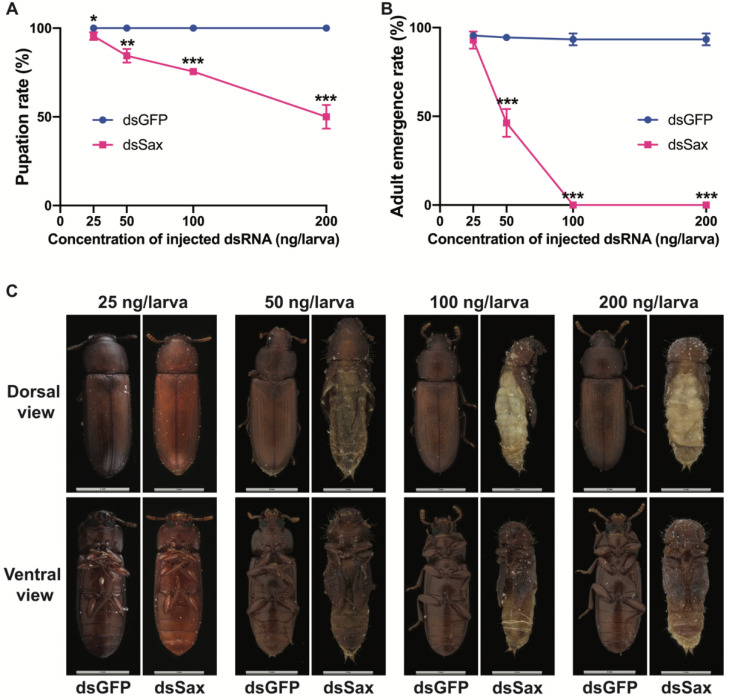
Effects of Sax RNAi on the pupation rate, adult emergence rate and phenotype. (**A**) Pupation rate in different doses of dsGFP and dsSax-treated groups. (**B**) Adult eclosion rate in different doses of dsGFP and dsSax-treated groups. (**C**) The phenotype of adults or failed emergence of adults from different doses of dsGFP and dsSax groups. Asterisks above bars indicate significant differences between the treatment and corresponding control, * *p* < 0.05, ** *p* < 0.01, *** *p* < 0.001 by *t*-test.

**Figure 6 molecules-27-06017-f006:**
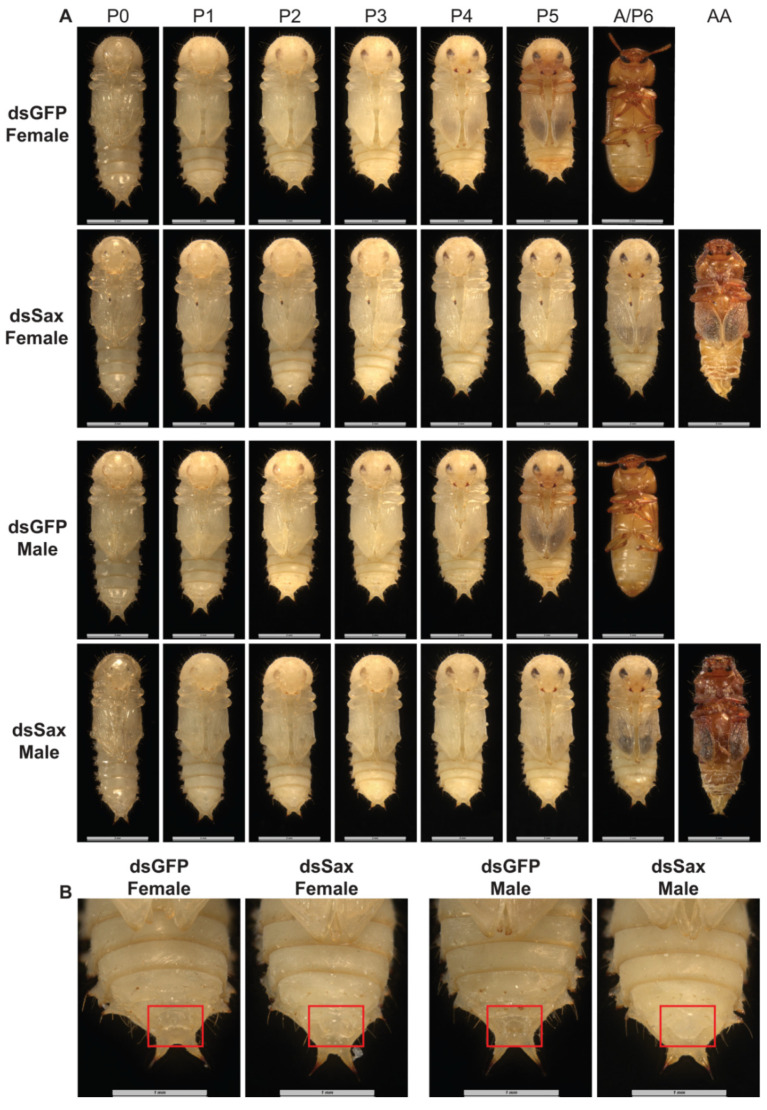
The ventral view of newly formed female and male pupae and adults from the dsGFP (100 ng/larva) and dsSax (100 ng/larva) groups. (**A**) The ventral view of newly formed female and male pupae and adults from the dsGFP and dsSax groups (P0: newly formed pupa; P1: 1-day pupa; P2: 2-day pupa; P3: 3-day pupa; P4: 4-day pupa; P5: 5-day pupa; P6: 6-day pupa; A: adult; AA: abnormal adult). (**B**) The ventral view of the enlarged female and male pupal abdomen in the dsGFP and dsSax treated groups. The red boxes indicate the genitals.

**Figure 7 molecules-27-06017-f007:**
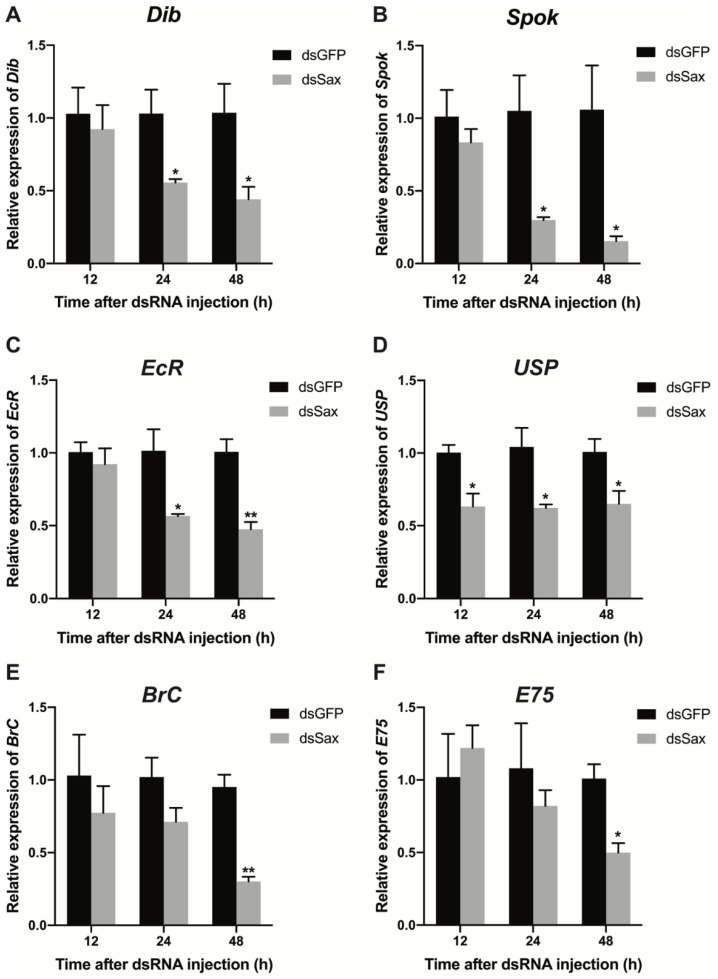
The effects of Sax RNAi on the expression of 20E pathway genes. (**A,B**) Relative expression levels of 20E biosynthesis genes *Dib* and *Spok* 12, 24 and 48 h post dsGFP (100 ng/larva) or dsSax (100 ng/larva) injection. (**C,D**) Relative expression levels of 20E receptor genes *EcR* and *USP* 12, 24 and 48 h post dsGFP (100 ng/larva) or dsSax (100 ng/larva) injection. (**E,F**) Relative expression levels of 20E downstream genes *BrC* and *E75* 12, 24 and 48 h post dsGFP (100 ng/larva) or dsSax (100 ng/larva) injection. Asterisks above bars indicate significant differences between the treatment and corresponding control, * *p* < 0.05, ** *p* < 0.01 by *t*-test.

**Figure 8 molecules-27-06017-f008:**
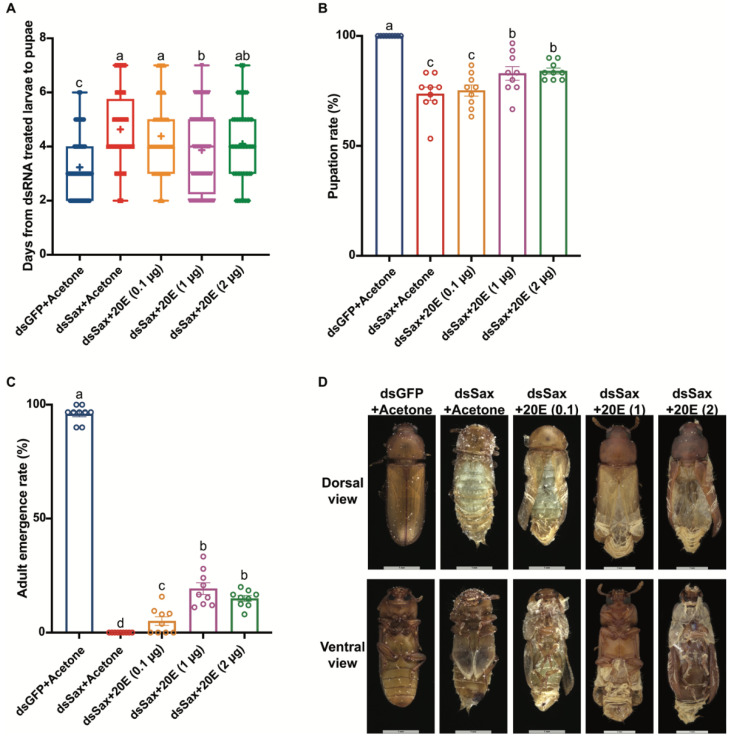
Effects of 20E application on the larval-pupal-adult development of *T. castaneum* post Sax RNAi. (**A**) Days from dsRNA treated 19-day larvae to newly formed pupae from the acetone or different doses of 20E (0.1, 1 and 2 μg/larva) applicated groups 24 h post 100 ng/larva dsGFP or dsSax treatment. (**B**) Pupation rate in the acetone or different doses of 20E (0.1, 1 and 2 μg/larva) applicated groups 24 h post 100 ng/larva dsGFP or dsSax treatment. (**C**) Adult emergence rate in the acetone or different doses of 20E (0.1, 1 and 2 μg/larva) applicated groups 24 h post 100 ng/larva dsGFP or dsSax treatment. (**D**) The phenotype of adults or failed emergence of adults from acetone or different doses of 20E (0.1, 1 and 2 μg/larva) applicated groups 24 h post 100 ng/larva dsGFP or dsSax treatment. Different letters on the bars indicate that the means ± SEM are significantly different among treatments by *t*-test.

## Data Availability

Data are contained within the article and Appendix A.

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
