# Peer review of "The TGF-β Receptor Gene Saxophone Influences Larval-Pupal-Adult Development in Tribolium castaneum"

_molecules, 2022, doi:10.3390/molecules27186017_

Round 1
Reviewer 1 Report
This is a well-written article in which the authors investigate the role of the type I TGF-β receptor Saxophone (Sax) in mediating development in red flour beetle, Tribolium castaneum and find that TGF-β receptor gene Sax influences development via 20E signalling in T. castaneum. Through the phylogenetic analysis, they confirmed that TcSax is a TGF-βRI.
The manuscript is interesting and worth publication, however, I have some observations:
1. Section 2.4: The method used to measure gene expression (2^delta delta Ct) assumes 100% amplification efficiency; please include in Table 1 the amplification efficiency, Tm, and product size for all primers sets.
2. Replace figure 2 with a higher resolution one and enter the bootstrap numbers.
3. Insert asterisks to indicate the significant differences in figure 3 and NS in the figures where it has not been indicated.
Author Response
We appreciate the reviewer for the important comment. We added the amplification efficiency, Tm, and product size for all primers set in section 2.4. According to the suggestion of reviewer 2, we moved Table 1 to supplementary materials as Table S1. And we used a higher resolution picture and entered the bootstrap value in Figure 2. For Figure 3, we added the significant differences.
Reviewer 2 Report
Summary
In the manuscript “The TGF-β receptor gene saxophone (sax) influences larval-pupal-adult development in Tribolium castaneum” Li et al., described the function of the type I TGFβ receptor Saxophone and its interaction with 20E signaling. The authors compared the sequence of sax from T. castaneum with other insects, tested the function of sax using RNAi, and concluded that the level of sax has a direct correlation with pupation and eclosion rate of floor beetles. Knocking down ecdysone signaling receptor EcR and USP gave similar results to that of sax. The authors linked the function of sax with 20E biosynthesis genes, receptors, and target genes using qPCR and where the addition of 20E partially rescued the sax RNAi phenotypes. Overall, the hypothesis the authors made is supported by the data provided, along with a well-described introduction and method section. There are, however, a few points the authors can improve on which are mentioned below.
Major issues
One of the experimental data I find missing in the present article and could improve the manuscript is the spatial expression profile of sax in T. castaneum. If the lab has access to in-situ hybridization technology the authors can generate spatial expression data of sax and its target genes in WT and RNAi individuals. This would be particularly helpful in explaining certain phenotypes such as abnormal eye development in dsSax individual in Figure 4D and enlarged/abnormal genitalia in Figure 6B.
The authors have mentioned that they performed the qPCR experiments with three biological and three technical replicates. In our lab, we have observed bias with three biological replicates and hence currently we try to do at least five biological replicates to confirm our results. I am worried that some of the data such as in Figure S5E might get biased due to fewer replicates and also since the paper heavily uses the qPCR data for drawing the link between sax and 20E signaling. However, if three replicates provide standard results in T. castaneum and have been a standard in past experiments I would not recommend any additional data.
Section 2.8: What does the author mean by topical application? Does the 20E was applied on the surface of the beetles or injected into the hemolymph with the needle? Please clarify this point.
The discussion section can be better. For example:
In Line 292-294 the authors mentioned reduction in sax results in the change of Dib and Spok, and not phantom and shadow. Why is this so?
The authors can add a summary illustration in the discussion section describing all the interactions with sax and 20E genes. This will be very helpful to get the essence of the manuscript.
The broader implications of the study and the limitations need to be highlighted better.
Also, the discussion section currently looks like one point after another from the result section. The authors can rearrange the section to provide a better flow with each paragraph describing one part of the points described in the first paragraph.
Minor issues
1) The authors can start a paragraph with a proper topic sentence that can describe the rest of the paragraph. For example, in the second paragraph of the introduction, the first sentence is on the function of TGFβ in vertebrates, while the rest of the paragraph describes TGFβ function in invertebrates. Sentence three in the paragraph is a better topic sentence.
2) Correct punctuations. There are multiple errors throughout the manuscript.
3) Authors should use a proper naming scheme for genes and proteins. Genes should be italicized with a small first letter and protein with normal and capital first letter.
4) Line 27: Provide the full form of EcR and USP.
5) Line 32: Remove underline
6) Line 48: Add the full form of POU and TFAM.
7) Line 50-52: Add the species where this study was done.
8) Line 70: Thick veins
9) Line 77-78: All the studies mentioned before are on insects (invertebrates). Why are the authors relating the role of sax in other vertebrates and connecting with the hypothesis?
10) Line 80: remove underline
11) Line 126: The authors can move the table to the supplementary file.
12) Line 150: , and adult eclosion rate.
13) Figure 6B. The enlarged abdomen is not properly visible at the current magnification. Can the authors provide a high magnification image of the region of interest with better contrast?
Author Response
We appreciated your constructive comments. Please see the attachment.

Round 2
Reviewer 2 Report
The authors have addressed all the comments.